# Neural Operator Variational Inference based on Regularized Stein Discrepancy for Deep Gaussian Processes

## Abstract

A Deep Gaussian Process (DGP) model is a hierarchical composition of GP models that provides a deep Bayesian nonparametric approach to infer the posterior. Exact Bayesian inference is usually intractable for DGPs, motivating the use of various approximations. We theoretically demonstrate that the traditional alternative of mean-field Gaussian assumptions across the hierarchy leads to lack of expressiveness and efficacy of DGP models, whilst stochastic approximation often incurs a significant computational cost. To address this issue, we propose **N**eural **O**perator **V**ariational **I**nference (NOVI) for Deep Gaussian Processes, where a sampler is obtained from a neural generator through minimizing Regularized Stein Discrepancy in $\mathcal{L}_2$ space between the approximate distribution and true posterior. Wherein, a minimax problem is obtained and solved by Monte Carlo estimation and subsampling stochastic optimization techniques. We experimentally demonstrate the effectiveness and efficiency of the proposed model, by applying it to a more flexible and wider class of posterior approximations on data ranging in size from hundreds to tens of thousands. By comparison, NOVI is superior to previous methods in both classification and regression.

## 1 Introduction

Gaussian processes (GPs) Rasmussen & Williams (2006) have proven to be extraordinarily effective as a tool for statistical inference and machine learning, for example when combined with thresholding to perform classification tasks via probit models Rasmussen & Williams (2006); Neal (1997) or to find interfaces in Bayesian inversion Iglesias et al. (2016). However, the joint Gaussian assumption of the latent function values can be restrictive in a number of circumstancesDutordoir et al. (2021). This is due to at least two factors: first, not all prior information is purely expressible in terms of mean and covariance, and second, Gaussian marginals are insufficient for many applications such as in the sparse data scenario, where the constructed probability distribution is far from posterior contraction. Thus, Deep Gaussian processes (DGPs) Damianou & Lawrence (2013) have been proposed to circumvent both of these constraints.

A DGP model is a hierarchical composition of GP models that provides a deep probabilistic nonparametric approach with sound uncertity quantification Ober & Aitchison (2021). The non-Gaussian distribution over composition functions yields both expressive capacity and intractable inference Dunlop et al. (2018). Previous work on DGP models utilized variational inference with a combination of sparse Gaussian processes Snelson & Gharahmani (2005); Quiñonero-Candela & Rasmussen (2005) and mean-field Gaussian assumptions Hensman et al. (2015); Deisenroth & Ng (2015); Gal et al. (2014); Hensman et al. (2013); Hoang et al. (2015; 2016); Titsias (2009b) for approximate posterior adjoint with stochastic optimization to scale up DGPs to large datasets like DSVI Salimbeni & Deisenroth (2017). These strategies often incorporate a collection of inducing points ($M \ll N$) whose position is learned alongside the other model hyperparameters, reduicng the training cost to $\mathcal{O}\left(NM^2\right)$.

While the mean-field Gaussian assumptions of the approximate posterior simplifies the computation, these assumptions impose overly stringent constraints, potentially limiting the expressiveness and effectiveness of such deterministic approximation approaches for DGP models Havasi et al. (2018);

Yu et al. (2019); Ustyuzhaninov et al. (2020); Lindinger et al. (2020). To solve the aforementioned problems, SGHMC Havasi et al. (2018) draws unbiased samples from the posterior belief using the stochastic approximation approach. However, due to its sequential sampling method, generating such samples is computationally expensive for both training and prediction, and its convergence is more challenging to evaluate in finite time Gao et al. (2021). Despite previous literatureYu et al. (2019); Lindinger et al. (2020) has discussed such issues, they all used different variants of the same KL-divergence-based variational bound, which is not symmetric and usually not stable for optimizationGoodfellow et al. (2016); Huggins et al. (2018). In order to solve the above problems, we address the issue by operator variational inference Ranganath et al. (2016), a Stein discrepancy based black-box algorithm that uses operators for optimizing any operator objective with data subsampling, where a minimax problem is obtained and solved by Monte Carlo estimation.

The main contributions are as follows:

- We propose NOVI for DGPs, a novel variational framework based on Stein discrepancy and operator variational inference with a neural generator. It minimizes Regularized Stein Discrepancy in $\mathcal{L}_2$ space between the approximate distribution and true posterior to construct a more flexible and wider class of posterior approximations overcoming previous limitations caused by mean-field Gaussian posterior assumptions and minimization of KL divergence.

- We theoretically demonstrate that our training schedule is equivalent to optimizing the Fisher divergence between the approximation and the true posterior while the bias raised by our method can be bounded by Fisher divergence in Section 5.

- We experimentally demonstrate the effectiveness and efficiency of the proposed model on 8 UCI regression datasets and image classification datasets including MNIST, Fashion-MNIST and CIFAR-10, which outperforms state-of-the-art approximation methods. Additional ablation study demonstrates that our method is superior in preventing overfitting.

Our code is publicly available at https://github.com/studying910/NOVI-DGP.

## 2 PRELIMINARY

### 2.1 GAUSSIAN PROCESSES

Let a random function $f : \mathbb{R}^D \to \mathbb{R}$ map $N$ training inputs $\boldsymbol{X} = \{\boldsymbol{x}_n\}_{n=1}^N$ to a collection of noisy observed outputs $\boldsymbol{y} = \{y_n\}_{n=1}^N$. In general, a zero mean GP prior is imposed on the function $f$, i.e., $f \sim \mathcal{GP}(0, k)$ where $k$ represents a covariance function $k : \mathbb{R}^D \times \mathbb{R}^D \to \mathbb{R}$. Let $\mathbf{f} = \{f(\boldsymbol{x}_n)\}_{n=1}^N$ represent the latent function values at the inputs $\boldsymbol{X}$. This assumption yields a multivariate Gaussian prior over the function values $p(\mathbf{f}) = \mathcal{N}(\mathbf{f}|0, \boldsymbol{K_{XX}})$ where $[\boldsymbol{K_{XX}}]_{ij} = k(\boldsymbol{x}_i, \boldsymbol{x}_j)$. In this work, we suppose $\boldsymbol{y}$ is contaminated by an i.i.d noise, thus $p(\boldsymbol{y}|\mathbf{f}) = \mathcal{N}(\boldsymbol{y}|\mathbf{f}, \sigma^2 \boldsymbol{I})$ where $\sigma^2$ is the noise variance. The GP posterior of the latent output $p(\mathbf{f}|\boldsymbol{y})$ has a closed-form solution Rasmussen & Williams (2006) but suffers from $\mathcal{O}(N^3)$ computational cost and $\mathcal{O}(N^2)$ storage requirement, thus limiting its scalability to big data.

Advanced sparse methods have been developed to set so-called *inducing points* $\boldsymbol{Z} = \{\boldsymbol{z}_m\}_{m=1}^M$ ($M \ll N$) from the input space and the associated inducing outputs known as *inducing variables*: $\mathbf{u} = \{\mathbf{u}_m = f(\boldsymbol{z}_m)\}_{m=1}^M$ Titsias (2009a); Snelson & Gharahmani (2005); Quiñonero-Candela & Rasmussen (2005), with a time complexity of $\mathcal{O}(NM^2)$. In this *Sparse GPs* (SGPs) paradigm, *inducing variables* $\mathbf{u}$ share a joint multivariate Gaussian distribution with $\mathbf{f}$: $p(\mathbf{f}, \mathbf{u}) = p(\mathbf{f}|\mathbf{u})p(\mathbf{u})$ where the condition is specified as:

$$p(\mathbf{f}|\mathbf{u}) = \mathcal{N}(\boldsymbol{K_{XZ}K_{ZZ}^{-1}}\mathbf{u}, \boldsymbol{K_{XX}} - \boldsymbol{K_{XZ}K_{ZZ}^{-1}K_{ZX}}) \tag{1}$$

and $p(\mathbf{u}) = \mathcal{N}(\mathbf{u}|0, \boldsymbol{K_{ZZ}})$ is the prior over the inducing outputs.

To solve the intractable posterior distribution of inducing variables $p(\mathbf{u}|\mathbf{y})$, *Sparse variational GPs* (SVGPs) Titsias (2009a); Hensman et al. (2015) reformulate the posterior inference problem as variational inference (VI) and confine the variational distribution to be $q(\mathbf{f}, \mathbf{u}) = p(\mathbf{f}|\mathbf{u})q(\mathbf{u})$ Hensman et al. (2013); Titsias (2009a); Gal et al. (2014); Salimbeni & Deisenroth (2017). This method approximates $q(\mathbf{u}) = \mathcal{N}(\boldsymbol{m}, \boldsymbol{S})$ Hensman et al. (2015); Deisenroth & Ng (2015); Gal et al.

(2014); Hensman et al. (2013); Hoang et al. (2015; 2016); Titsias (2009b), then a Gaussian marginal[1]. is obtained by maximizing the evidence lower bound (ELBO) Hoffman et al. (2013).

## 2.2 DEEP GAUSSIAN PROCESSES

A multi-layer DGP model is a hierarchical composition of GP models constructed by stacking the muti-output SGPs together Damianou & Lawrence (2013). Consider a model with $L$ layers and $D_l$ independent random functions in layer $\ell = 1, \ldots, L$ such that output of the $\ell$-1-th layer $\mathbf{F}_{\ell-1}$ is used as an input to the $\ell$-th layer, i.e., $\mathbf{F}_\ell = \{\mathbf{f}_{\ell,1} = f_{\ell,1}(\mathbf{F}_{\ell-1}), \cdots, \mathbf{f}_{\ell,D_\ell} = f_{\ell,D_\ell}(\mathbf{F}_{\ell-1})\}$, where $f_{\ell,d} \sim \mathcal{GP}(0, k_\ell)$ for $d = 1, \ldots, D_\ell$ and $\mathbf{F}_0 \triangleq \boldsymbol{X}$. The inducing points and corresponding inducing variables for DGP layers are denoted by $\mathsf{Z} = \{\boldsymbol{Z}_\ell\}_{\ell=1}^L$ and $\mathsf{U} = \{\mathbf{U}_\ell\}_{\ell=1}^L$ respectively where $\mathbf{U}_\ell = \{\mathbf{u}_{\ell,1} = f_{\ell,1}(\boldsymbol{Z}_\ell), \cdots, \mathbf{u}_{\ell,D_\ell} = f_{\ell,D_\ell}(\boldsymbol{Z}_\ell)\}$. Let $\mathsf{F} = \{\mathbf{F}_\ell\}_{\ell=1}^L$, the DGP model design yields the following joint model density:

$$p(\boldsymbol{y}, \mathsf{F}, \mathsf{U}) = p(\boldsymbol{y}|\mathbf{F}_L) \prod_{\ell=1}^L p(\mathbf{F}_\ell|\mathbf{F}_{\ell-1}, \mathbf{U}_\ell) p(\mathsf{U}). \tag{2}$$

Here we place independent GP priors within and across layers on $\mathsf{U}$: $p(\mathsf{U}) = \prod_{l=1}^L p(\mathbf{U}_l) = \prod_{l=1}^L \prod_{d=1}^{D_\ell} \mathcal{N}(\mathbf{u}_{\ell,d}|0, \boldsymbol{K}_{\boldsymbol{Z}_\ell \boldsymbol{Z}_\ell})$ and in the same way as Equation (1), the condition is defined as:

$$p(\mathbf{F}_\ell|\mathbf{F}_{\ell-1}, \mathbf{U}_\ell) = \prod_{d=1}^{D_\ell} \mathcal{N}(\mathbf{f}_{\ell,d}|\boldsymbol{K}_{\mathbf{F}_{\ell-1}\boldsymbol{Z}_\ell}\boldsymbol{K}_{\boldsymbol{Z}_\ell\boldsymbol{Z}_\ell}^{-1}\mathbf{u}_{\ell,d}, \boldsymbol{K}_{\mathbf{F}_{\ell-1}\mathbf{F}_{\ell-1}} - \boldsymbol{K}_{\mathbf{F}_{\ell-1}\boldsymbol{Z}_\ell}\boldsymbol{K}_{\boldsymbol{Z}_\ell\boldsymbol{Z}_\ell}^{-1}\boldsymbol{K}_{\boldsymbol{Z}_\ell\mathbf{F}_{\ell-1}}). \tag{3}$$

As an extension of Variational Inference with DGPs, DSVI Salimbeni & Deisenroth (2017) approximates the posterior by requiring the distribution across the inducing outputs to be a-posteriori Gaussian and independent amongst distinct GPs to obtain an analytical ELBO (known as the mean-field assumption Opper & Saad (2001); Hoffman et al. (2013), $q(\mathbf{u}_{\ell,1:D_\ell}) = \mathcal{N}(\boldsymbol{m}_{\ell,1:D_\ell}, \boldsymbol{S}_{\ell,1:D_\ell})$, where $\boldsymbol{m}_{\ell,1:D_\ell}$ and $\boldsymbol{S}_{\ell,1:D_\ell}$ are variational parameters. By iteratively sampling the layer outputs and utilizing the reparameterisation trick Kingma & Welling (2013), DSVI enables scalability to big datasets.

As mentioned in Section 1, while the mean-field Gaussian assumptions of the variational posterior $q(\mathcal{U})$ makes it simple to analytically marginalise out the inducing outputs, these assumptions impose overly stringent constraints, potentially limiting the expressiveness and effectiveness of such deterministic approximation approaches for DGP models. In particular, according to Bayes' Rule, the true posterior distribution can be written as:

$$p(\mathsf{U}|\boldsymbol{y}) = \frac{p(\mathsf{U}) p(\boldsymbol{y}|\mathsf{U})}{p(\boldsymbol{y})} = \frac{\int p(\boldsymbol{y}, \mathsf{F}, \mathsf{U}) d\mathsf{F}}{p(\boldsymbol{y})} \tag{4}$$

Due to the fact that the latent functions $\mathbf{F}_1, \cdots, \mathbf{F}_{L-1}$ are inputs to the non-linear kernel function, the likelihood term $p(\boldsymbol{y}|\mathsf{U})$ in Equation (4) is intractable and $p(\mathsf{U}|\boldsymbol{y})$ is often non-Gaussian in reality. Moreover, the KL-based optimization often leads to unstable training Huggins et al. (2018).

To address this issue, we present a new variational family that provides both efficient computation and expressiveness based on *Operator Variational Inference* (OVI) Ranganath et al. (2016), while simultaneously learning preservable transformations and generating unbiased posterior samples constructed by neural networks, as detailed in Section 3 and Section 4.

## 3 OVI AND STEIN DISCREPANCY

**Definition 1.** *Let $p(\boldsymbol{x})$ be a probability density supported on $\mathcal{X} \subseteq \mathbb{R}^d$ and $\boldsymbol{\phi} : \mathcal{X} \to \mathbb{R}^d$ be a differentiable function, we define Langevin-Stein Operator (LSO) Ranganath et al. (2016) as:*

$$\mathcal{A}_p\boldsymbol{\phi}(\boldsymbol{x}) \triangleq \nabla_{\boldsymbol{x}} \log p(\boldsymbol{x})^T \boldsymbol{\phi}(\boldsymbol{x}) + \mathrm{Tr}(\nabla_{\boldsymbol{x}}\boldsymbol{\phi}(\boldsymbol{x})). \tag{5}$$

**Definition 2.** *(Stein's Discrepancy) Hu et al. (2018); Grathwohl et al. (2020); di Langosco et al. (2021) Let $p(\boldsymbol{x})$, $q(\boldsymbol{x})$ be probability densities supported on $\mathcal{X} \subseteq \mathbb{R}^d$. Stein discrepancy is defined by considering the maximum violation of Stein identity for $\boldsymbol{\phi}$ in some proper function set $\mathcal{F}$*

$$\mathcal{S}(q, p) \triangleq \sup_{\boldsymbol{\phi} \in \mathcal{F}} \mathbb{E}_{\boldsymbol{x} \sim q}[\mathcal{A}_p\boldsymbol{\phi}(\boldsymbol{x})]. \tag{6}$$

---

[1]The solution is given in App. A

Like previous methods Hu et al. (2018); Grathwohl et al. (2020), we take the function space $\mathcal{F}$ in Stein discrepancy (6) to be the $\mathcal{L}_2$ space and parameterize $\phi$ with a neural network $\phi_{\boldsymbol{\eta}}$ as a discriminator

$$\text{LSD}\left(q, p; \boldsymbol{\eta}\right) \triangleq \max_{\boldsymbol{\eta}}\{\mathbb{E}_{\boldsymbol{x} \sim q}[\nabla_{\boldsymbol{x}} \log p\left(\boldsymbol{x}\right)^T \phi_{\boldsymbol{\eta}}\left(\boldsymbol{x}\right) + \text{Tr}\left(\nabla_{\boldsymbol{x}} \phi_{\boldsymbol{\eta}}\left(\boldsymbol{x}\right)\right)]\}. \tag{7}$$

which is referred to the Learned Stein Discrepancy (LSD) Grathwohl et al. (2020). Neural networks as functions are not by definition square integrable, as they do not by default disappear at infinity. In order to satisfy the conditions of Stein's identityLiu & Wang (2016), an $\mathcal{L}_2$ regularizer with strength $\lambda \in \mathbb{R}^+$ is applied to LSD to gain a Regularized Stein Discrepancy (RSD)

$$\text{RSD}\left(q, p; \boldsymbol{\eta}\right) \triangleq \max_{\boldsymbol{\eta}}\{\mathbb{E}_{\boldsymbol{x} \sim q}[\nabla_{\boldsymbol{x}} \log p\left(\boldsymbol{x}\right)^T \phi_{\boldsymbol{\eta}}\left(\boldsymbol{x}\right) + \text{Tr}\left(\nabla_{\boldsymbol{x}} \phi_{\boldsymbol{\eta}}\left(\boldsymbol{x}\right)\right)] - \lambda \mathbb{E}_{\boldsymbol{x} \sim q}[\phi_{\boldsymbol{\eta}}(\boldsymbol{x})^T \phi_{\boldsymbol{\eta}}\left(\boldsymbol{x}\right)]\}. \tag{8}$$

In Bayesian posterior inference, we take $p$ and $q_{\boldsymbol{\theta}}$ as the true posterior and approximate posterior, respectively, where $\boldsymbol{\theta} \in \Theta$ and $\Theta$ is a set of variational parameters. Stein divergence in Equation (6) is usually used as an objective of OVI Ranganath et al. (2016), which is a black-box algorithm that uses operators for optimizing any operator objective with data subsampling and a wider class of posterior approximations that does not require a tractable density. Given parameterizations of the variational family $\Theta$ and the discriminator $\phi_{\boldsymbol{\eta}}$, OVI seeks to solve a minimax problem

$$\theta^\star = \arg \inf_{\boldsymbol{\theta} \in \Theta} \sup_{\boldsymbol{\eta}} \mathbb{E}_{\boldsymbol{x} \sim q_\theta}\left[\mathcal{A}_p \phi_{\boldsymbol{\eta}}\left(\boldsymbol{x}\right)\right]. \tag{9}$$

## 4 DEEP GAUSSIAN PROCESSES WITH NEURAL OPERATOR VARIATIONAL INFERENCE

Now we will discuss the algorithm design for the Bayesian inference problem of sampling the posterior $p(\mathbf{U}|\mathcal{D})$ for DGPs. For consistency, we continue to use notation in Section 2.2. Let $\mathcal{D} = \{\boldsymbol{x}_n, y_n\}_{n=1}^N$ represent the training dataset, $\mathbf{U} \triangleq \{\mathbf{U}_\ell\}_{\ell=1}^L$ represent inducing variables and $\boldsymbol{\nu}$ represent the DGP model hyperparameters including inducing points locations, kernel hyperparameters and noise variance.

### 4.1 NEURAL NETWORK AS GENERATOR

Let $q_0(\boldsymbol{\epsilon})$ be the reference distribution that generates noise $\boldsymbol{\epsilon} \in \mathbb{R}^{d_0}$. Let $g_{\boldsymbol{\theta}}$ represent our sampler, which is a black-box generator parameterized by a multi-layer neural network. Let $q_{\boldsymbol{\theta}}\left(\mathbf{U}\right)$ be the underlying density of the generated samples $\mathbf{U} = g_{\boldsymbol{\theta}}(\boldsymbol{\epsilon})$. In summary, our setup is as follows:

$$\boldsymbol{\epsilon} \sim q_0\left(\boldsymbol{\epsilon}\right), \qquad g_{\boldsymbol{\theta}}\left(\boldsymbol{\epsilon}\right) = \mathbf{U} \sim q_{\boldsymbol{\theta}}\left(\mathbf{U}\right)$$

Neural networks as a generator have a high capacity and can well approximate almost any distribution by transforming simple ones such as Gaussian or uniform distribution with many applications in deep generative models Huszár (2017); Mescheder et al. (2017); Titsias & Ruiz (2019); Cybenko (1989); Lu & Lu (2020); Perekrestenko et al. (2020); Yang et al. (2022). Since the generative distribution $q_{\boldsymbol{\theta}}\left(\mathbf{U}\right)$ is implict, KL divergence is not applicable as a measure between $q_{\boldsymbol{\theta}}\left(\mathbf{U}\right)$ and the true posterior $p(\mathbf{U}|\mathcal{D})$ in this case. Therefore, it is reasonable to use OVI and RSD to construct a better objective.

### 4.2 TRAINING SCHEDULE

In Section 3 we have reviewed OVI, a method using Langevin-Stein operator and allowing for a more flexible representation of the posterior geometry beyond the commonly used Gaussian distribution used in vanilla VI. We extend it to applications in inducing points posterior inference for DGP model by learning the parameters of neural network generator to best fit the data. Since our discriminator $\phi_{\boldsymbol{\eta}}$ is sufficiently expressive, we produce an objective whose expectation[2] is 0 if and only if the true posterior $p(\mathbf{U}|\mathcal{D})$ and the approximate distribution $q(\mathbf{U})$ are equivalent. During training, we will minimize

$$\mathcal{L}(\boldsymbol{\theta}, \boldsymbol{\nu}) = \text{RSD}\left(q_{\boldsymbol{\theta}}(\mathbf{U}), p(\mathbf{U}|\mathcal{D}, \boldsymbol{\nu}); \phi_{\boldsymbol{\eta}}\right) \tag{10}$$

---

[2]The expectation doesn't include the regularization term.

with respect to $\boldsymbol{\theta}$ and jointly optimise the model hyperparameters $\boldsymbol{\nu}$ by maximizing log-likelihood via Monte Carlo sampling. However, this procedure is difficult due to the supremum on r.h.s. of Equation (8). In order to obtain the optimized network parameters $\boldsymbol{\theta}$, we iteratively update the generator $g_{\boldsymbol{\theta}}$ and the discriminator $\phi_{\boldsymbol{\eta}}$ in an alternating manner where the discriminator is trained to more accurately estimate the Stein Discrepancy and the generator is trained to minimize the estimation of the discrepancy. The proposed training algorithm is summarized in Algorithm 1 which we refer to it as *Neural Operator Variational Inference* (NOVI) for DGP.

---

**Algorithm 1:** NOVI for DGP

---

**Input:** training data $\mathcal{D} = \{\boldsymbol{x}_n, y_n\}_{n=1}^N$, penalty parameter $\lambda$, $n_c$ number of iterations for training the critic, learning rate $\alpha, \beta, \gamma$, M batch size, sample number K
**Initialize** discriminator $\boldsymbol{\eta}$, generator $\boldsymbol{\theta}$, DGP hyperparameters $\boldsymbol{\nu}$
**repeat**
 **for** $j = 1$ **to** $n_c$ **do**
  Sample a minibatch $\{\boldsymbol{x}_i, y_i\}_{i=1}^M \sim \mathcal{D}$
  Generate i.i.d. noise inputs $\boldsymbol{\epsilon}_1 \ldots \boldsymbol{\epsilon}_K$ from $q_0$
  Obtain fake sample $g_{\boldsymbol{\theta}}(\boldsymbol{\epsilon}_1) \ldots g_{\boldsymbol{\theta}}(\boldsymbol{\epsilon}_K)$
  Compute empirical loss $\widehat{\mathrm{RSD}}(q_{\boldsymbol{\theta}}, p; \phi_{\boldsymbol{\eta}})$
  $\boldsymbol{\eta} \leftarrow \boldsymbol{\eta} - \alpha \nabla_{\eta} \widehat{\mathrm{RSD}}(q_{\boldsymbol{\theta}}, p; \phi_{\boldsymbol{\eta}})$
 **end for**
 Compute empirical loss $\widehat{\mathcal{L}}(\boldsymbol{\theta}, \boldsymbol{\nu})$
 $\boldsymbol{\theta} \leftarrow \boldsymbol{\theta} - \beta \nabla_{\boldsymbol{\theta}} \widehat{\mathcal{L}}(\boldsymbol{\theta}, \boldsymbol{\nu})$
 $\boldsymbol{\nu} \leftarrow \boldsymbol{\nu} - \gamma \frac{1}{K} \sum_{k=1}^K \nabla_{\boldsymbol{\nu}} \log p(\boldsymbol{y}, \mathbf{U}^k | \boldsymbol{\nu})$
**until** $\boldsymbol{\theta}, \boldsymbol{\nu}$ converge

---

In our implementation, we utilize Monte Carlo method to estimate the objective (10) and RSD(8):

$$\widehat{\mathrm{RSD}}(q_{\boldsymbol{\theta}}, p; \phi_{\boldsymbol{\eta}}) = \frac{1}{K} \sum_{k=1}^K \left( \nabla_{\mathbf{U}} \log p(\mathbf{U}|\mathcal{D}, \boldsymbol{\nu})^T \big|_{\mathbf{U}=\mathbf{U}^k} \phi_{\boldsymbol{\eta}}(\mathbf{U}^k) + \mathbb{E}_{\boldsymbol{\omega} \sim \mathcal{N}(0, \boldsymbol{I})}(\boldsymbol{\omega}^T \nabla_{\mathbf{U}} \phi_{\boldsymbol{\eta}}(\mathbf{U})|_{\mathbf{U}=\mathbf{U}^k} \boldsymbol{\omega}) \right)$$
$$- \lambda \frac{1}{K} \sum_{k=1}^K (\phi_{\boldsymbol{\eta}}(\mathbf{U}^k)^T \phi_{\boldsymbol{\eta}}(\mathbf{U}^k))$$
$$\widehat{\mathcal{L}}(\boldsymbol{\theta}, \boldsymbol{\nu}) = \widehat{\mathrm{RSD}}(q_{\boldsymbol{\theta}}, p; \phi_{\boldsymbol{\eta}^\star}),$$
(11)

where $\phi_{\boldsymbol{\eta}^\star}$ is the supremum of RSD estimate and the gradient with $\boldsymbol{\theta}$ and $\boldsymbol{\nu}$ is computed via automatic differentiation. We use Hutchinson estimator Hutchinson (1989) to compute the expensive divergence of $\phi_{\boldsymbol{\eta}}$ in Equation (11), which is a simple yet effective way to obtain a stochastic estimate of the trace of a matrix. It can reduce the time complexity from $\mathcal{O}(D^2)$ to $\mathcal{O}(D)$ where $D$ is the dimensionality of the matrix. In Theorem 1, we prove that the score function $\nabla_{\mathbf{U}} \log p(\mathbf{U}|\mathcal{D}, \boldsymbol{\nu})$ can be evaluated by Monte Carlo method, which shows that RSD can be utilized as a reasonable objective to update the parameters of the generator network.

**Theorem 1.** *The score function $\nabla_{\mathbf{U}} \log p(\mathbf{U}|\mathcal{D}, \boldsymbol{\nu})$ in Equation (11) can be evaluated by Monte Carlo sampling (detailed proof can be seen in App. B):*

$$\nabla_{\mathbf{U}} \log p(\mathbf{U}|\mathcal{D}, \boldsymbol{\nu}) \approx -(\boldsymbol{\Delta}_1, \ldots, \boldsymbol{\Delta}_\ell, \ldots, \boldsymbol{\Delta}_L) + \nabla_{\mathbf{U}} \log \sum_{s=1}^S p(\boldsymbol{y}|\widehat{\mathbf{F}}_L^{(s)})$$
(12)

*where* $\boldsymbol{\Delta}_\ell = (\boldsymbol{K}_{\boldsymbol{Z}_\ell \boldsymbol{Z}_\ell}^{-1} \mathbf{u}_{\ell,1}, ..., \boldsymbol{K}_{\boldsymbol{Z}_\ell \boldsymbol{Z}_\ell}^{-1} \mathbf{u}_{\ell,d}, ..., \boldsymbol{K}_{\boldsymbol{Z}_\ell \boldsymbol{Z}_\ell}^{-1} \mathbf{u}_{\ell,D_\ell})$ *and* $\widehat{\mathbf{f}}_{\ell,d}^{(s)} \sim$
$\mathcal{N}(\boldsymbol{K}_{\widehat{\mathbf{F}}_{\ell-1} \boldsymbol{Z}_\ell} \boldsymbol{K}_{\boldsymbol{Z}_\ell \boldsymbol{Z}_\ell}^{-1} \mathbf{u}_{\ell,d}, \boldsymbol{K}_{\widehat{\mathbf{F}}_{\ell-1} \widehat{\mathbf{F}}_{\ell-1}} - \boldsymbol{K}_{\widehat{\mathbf{F}}_{\ell-1} \boldsymbol{Z}_\ell} \boldsymbol{K}_{\boldsymbol{Z}_\ell \boldsymbol{Z}_\ell}^{-1} \boldsymbol{K}_{\boldsymbol{Z}_\ell \widehat{\mathbf{F}}_{\ell-1}})$ *for* $\ell = 1, \ldots, L$, $S$ *is the number of samples involved in estimation.*

## 4.3 PREDICTION

Let $\mathcal{D}^\star = \{\boldsymbol{x}_n^\star, y_n^\star\}_{n=1}^T$ be the test data, to predict its value, we sample from the optimized generator and convert the input locations $\boldsymbol{x}$ to the test location $\boldsymbol{x}^\star$ in formula. We denote the function values at the test location as $\mathbf{F}_\ell^\star$. To obtain the final layer density we use

$$q(\mathbf{F}_L^\star) = \int \prod_{\ell=1}^L \prod_{d=1}^{D_\ell} p(\mathbf{f}_{\ell,d}^\star | \mathbf{F}_{\ell-1}^\star, \mathbf{u}_{\ell,d}) q_{\boldsymbol{\theta}^\star}(\mathbf{u}_{\ell,d}) \, d\mathbf{F}_{\ell-1}^\star d\mathbf{u}_{\ell,d}$$
(13)

where $\boldsymbol{\theta}^{\star}$ is the optimal of the generator and the first term of the integral $p(\mathbf{f}_{\ell,d}^{\star}|\mathbf{F}_{\ell-1}^{\star}, \mathbf{u}_{\ell,d})$ is conditional Gaussian. We leverage this consequence to draw samples from $q\left(\mathbf{F}_{L}^{\star}\right)$, and further perform the sampling using re-parameterization trick Salimbeni & Deisenroth (2017); Rezende et al. (2014); Kingma et al. (2015). Specifically, we first sample $\boldsymbol{\epsilon}^{\ell} \sim \mathcal{N}(0, \boldsymbol{I}_{D^{\ell}})$ and $\mathbf{U} \sim q_{\boldsymbol{\theta}^{\star}}(\mathbf{U})$, then recursively draw the sampled variables $\widehat{\mathbf{f}}_{\ell,d}^{\star} \sim p(\mathbf{f}_{\ell,d}^{\star}|\widehat{\mathbf{F}}_{\ell-1}^{\star}, \mathbf{u}_{\ell,d})$ for $\ell = 1, \ldots, L$ as:

$$\widehat{\mathbf{f}}_{\ell,d}^{\star} = \boldsymbol{K}_{\widehat{\mathbf{F}}_{\ell-1}^{\star}\boldsymbol{Z}_{\ell}} \boldsymbol{K}_{\boldsymbol{Z}_{\ell}\boldsymbol{Z}_{\ell}}^{-1} \mathbf{u}_{\ell,d} + \boldsymbol{\epsilon}_{\ell} \odot \sqrt{\operatorname{diag}\left(\boldsymbol{K}_{\widehat{\mathbf{F}}_{\ell-1}^{\star}\widehat{\mathbf{F}}_{\ell-1}^{\star}} - \boldsymbol{K}_{\widehat{\mathbf{F}}_{\ell-1}^{\star}\boldsymbol{Z}_{\ell}} \boldsymbol{K}_{\boldsymbol{Z}_{\ell}\boldsymbol{Z}_{\ell}}^{-1} \boldsymbol{K}_{\boldsymbol{Z}_{\ell}\widehat{\mathbf{F}}_{\ell-1}^{\star}}\right)}, \quad (14)$$

where the square root is element-wise. We define $\mathbf{F}_{0}^{\star} \triangleq \boldsymbol{X}^{\star}$ for the first layer and use $\operatorname{diag}\left(\cdot\right)$ to denote the vector of diagonal elements of a matrix. The diagonal approximation in Equation (14) holds since in DGP model, the $i$-th marginal of approximate posterior $q(\mathbf{f}_{(\ell,d)[i]})$ depends only on the corresponding inputs $\boldsymbol{x}_{i}$ Quiñonero-Candela & Rasmussen (2005). In our experiment, we concatenate $\boldsymbol{Z}_{\ell}$ and $\boldsymbol{\epsilon}$ to generate $\mathbf{U}$ to avoid overfitting Yu et al. (2019).

## 5 CONVERGENCE GUARANTEES

**Definition 3.** *The Fisher divergence Sriperumbudur et al. (2017) between two suitably smooth density functions is defined as*

$$F\left(q, p\right) = \int_{\mathbb{R}^{d}} \|\nabla \log q\left(\boldsymbol{x}\right) - \nabla \log p\left(\boldsymbol{x}\right)\|_{2}^{2} q\left(\boldsymbol{x}\right) d\boldsymbol{x}.$$

**Theorem 2.** *Training the generator with the optimal discriminator corresponds to minimizing the fisher divergence between $p_{\theta}$ and $q$. The corresponding optimal loss is (detailed proof can be seen in App. C)*

$$\mathcal{L}\left(\boldsymbol{\theta}, \boldsymbol{\nu}\right) = \frac{1}{4\lambda} F\left(q_{\boldsymbol{\theta}}\left(\mathbf{U}\right), p\left(\mathbf{U}|\mathcal{D}, \boldsymbol{\nu}\right)\right)$$

**Theorem 3.** *The bias of the estimation for prediction $\widehat{\mathbf{F}}_{L}^{\star}$ in Equation (14) from the DGPs exact evaluation can be bounded by the square root of the Fisher divergence between $q_{\boldsymbol{\theta}}(\mathbf{U})$ and $p\left(\mathbf{U}|\mathcal{D}, \boldsymbol{\nu}\right)$ up to multiplying a constant. (detailed proof can be seen in App. C)*

Theorem2 shows our algorithm is equivalent to minimizing Fisher divergence while Theorem3 guarantees a bounded bias of the estimation for prediction. Fisher divergence has proved useful in a variety of statistics and machine learning applications Huggins et al. (2018); Holmes & Walker (2017); Walker (2016). Connections between Fisher divergence and certain "rates of change" in KL divergence can be seen in de Bruijn's identity Barron (1986); Stam (1959) and Stein's identity Liu & Wang (2016); Park et al. (2012). Under mild conditions, according to Sobolev inequality, Fisher divergence is a stronger distance than KL divergence. In fact, it is stronger than a lot of other distances between distributions, such as total variation Chambolle (2004), Hellinger distance Beran (1977), Wasserstein distance Vallender (1974), etc Ley & Swan (2013).Huggins et al. (2018)showed that a suitable Fisher divergence upper-bounds Wasserstein distance, which suggests that approximations based on minimizing the former, compared to KL divergence, would lead to improved moment estimates.

## 6 RELATED WORKS

**OVI and Stein Discrepancies** Our method about the inference is inspired by OVI Ranganath et al. (2016) and Stein Neural SamplerHu et al. (2018) but the distinguish is ours concentrates on the DGP posterior and develop specific algorithms. Different from the general Bayesian model, the likelihood function of DGPs is not explicit, so we propose the stochastic gradient and Monte Carlo sampling method to calculate the score function (Theorem 1). OVI Ranganath et al. (2016) introduces an objective for inference similar to RSD but utilize a different class of discriminator and neither of the two methods Ranganath et al. (2016); Hu et al. (2018) applies many of state-of-the-art techniques we used for scalability such as Hutchinson estimator Hutchinson (1989).

**Variational Inference** Among the methods to address the limitations of mean-field variational inference, we can find methods that shares the same motivations as ours for DGPs including adding

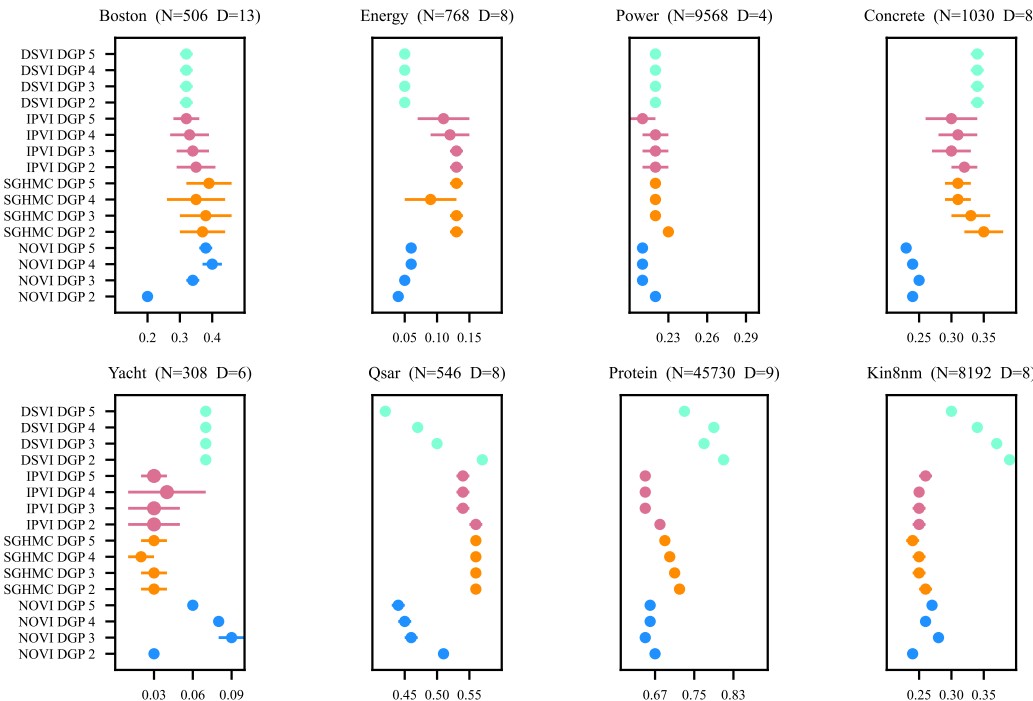

Figure 1: Regression mean test RMSE results by our NOVI method (blue), SGHMC (orange), IPVI(pink) and DSVI (cyan) for DGPs on UCI benchmark datasets. Lower is better. The mean is shown with error bars of one standard error.

dependencies among the latent variables using a structured variational familyLindinger et al. (2020), implicit distributions variational inference (IPVI)Mescheder et al. (2017); Ma et al. (2019); Yu et al. (2019); Sun et al. (2019), and normalizing flowsRezende & Mohamed (2015); Yu et al. (2021). Structured VILindinger et al. (2020) tends to design hand-crafted or expert solutions while the invertibility condition posts lots of constraints on the transformation form of normalizing flowsDinh et al. (2016). The main issue of IPVI is density ratio estimation, which is often addressed using adversarial networks. Controlling the bias and variance in density ratio estimation, however, is in general a very difficult problem, especially in high dimensional settingsSugiyama et al. (2012); Titsias & Ruiz (2019). In comparison, NOVI is more robust because it provides more accurate moment estimatesHuggins et al. (2018) than KL-based models, makes it feasible to handle higher dimensional latent variable distributionsGrathwohl et al. (2020) and reaches convergence faster.

# 7 EXPERIMENTS

We empirically evaluate and compare the performance of our method with Doubly Stochastic VI (DSVI) Salimbeni & Deisenroth (2017) for DGPs, which is implemented as our baseline model, Implicit Posterior VI (IPVI) for DGPs, which also constructs a neural network to model a posterior for approximate inference Yu et al. (2019) and state-of-the-art SGHMC model Havasi et al. (2018) using real-world datasets in regression and classification tasks both in small and large data regimes. All our experiments were run with exactly the same hyper-parameters and initializations. Detailed training information can be seen in App. E.

## 7.1 UCI REGRESSION BENCHMARK

Our experiments are conducted on 8 UCI regression datasets with size ranging from 308 to 45730. The performance metric is average RMSE of the test data. The results are shown in Figure 1 (tabular version can be seen in App. D.3). On four of the eight datasets, simply using 2-layer NOVI model achieves the best result and a huge performance gap against other three methods. On larger datasets,

| Data Set | Model | Time (L=3) | Iter(L=3) | Acc (L=3) | Time (L=4) | Iter (L=4) | Acc (L=4) |
|---|---|---|---|---|---|---|---|
| | DSVI | 0.34s/iter | 20K | - | 0.54s/iter | 20K | 97.41 |
| MNIST | IPVI | 0.49s/iter | 20K | - | 0.62s/iter | 20K | 97.80 |
| | SGHMC | 1.14s/iter | 20K | - | 1.22s/iter | 20K | 97.55 |
| | NOVI (ours) | 0.38s/iter | 10K | **97.94** | 0.50s/iter | 10K | **98.01** |
| | DSVI | 0.34s/iter | 20K | - | 0.50s/iter | 20K | 87.99 |
| Fashion-MNIST | IPVI | 0.48s/iter | 20K | - | 0.61s/iter | 20K | 88.90 |
| | SGHMC | 1.21s/iter | 20K | - | 1.25s/iter | 20K | 87.08 |
| | NOVI (ours) | 0.40s/iter | 10K | **88.96** | 0.55s/iter | 10K | **89.15** |
| | DSVI | 0.43s/iter | 20K | - | 0.66s/iter | 20K | 51.79 |
| CIFAR-10 | IPVI | 0.62s/iter | 20K | - | 0.78s/iter | 20K | 53.27 |
| | SGHMC | 8.04s/iter | 20K | - | 8.61s/iter | 20K | 52.81 |
| | NOVI (ours) | 0.43s/iter | 10K | **53.32** | 0.52s/iter | 10K | **53.42** |

Table 1: Mean test accuracy (%) and training details achieved by DSVI, SGHMC and NOVI (ours) DGP model for three image classification datasets. Batch size is set to 256 for all methods. L denotes the number of hidden layers. Our proposed method can also be combined with convolution kernels Kumar et al. (2018) to obtain a better result, for a fair comparison, we have not implemented here.

| Type | DSVI 2 | DSVI 3 | DSVI 4 | DSVI 5 |
|---|---|---|---|---|
| Time (s) | 0.835 | 0.903 | 0.965 | 1.339 |
| Iteration | 20K | 20K | 20K | 20K |
| Type | IPVI 2 | IPVI 3 | IPVI 4 | IPVI 5 |
| Time (s) | 0.117 | 0.162 | 0.211 | 0.260 |
| Iteration | 20K | 20K | 20K | 20K |
| Type | SGHMC 2 | SGHMC 3 | SGHMC 4 | SGHMC 5 |
| Time (s) | 0.630 | 1.000 | 1.490 | 1.870 |
| Iteration | 20K | 20K | 20K | 20K |
| Type | NOVI 2 | NOVI 3 | NOVI 4 | NOVI 5 |
| Time (s) | 0.391 | 0.613 | 0.863 | 1.123 |
| Iteration | 500 | 500 | 500 | 500 |

Table 2: Comparison of training time (s) of a single iteration and total training iterations on Energy dataset. Batch size is set to 1000 for all three methods. $*$ indicates that although IPVI takes less time per iteration, it requires a larger training iteration to converge, which is more time-consuming than our method.

like 'Power', 'Concrete', 'Qsar' and 'Protein', the deepest NOVI model outperforms other methods. We attribute this phenomenon to the overfitting of the deep model on small data sets. Additional results for real-world regression datasets can be seen in App. D.5.

## 7.2 IMAGE CLASSIFICATION

We apply our method to MNIST LeCun et al. (1998), Fashion-MNIST Xiao et al. (2017) and CIFAR-10 Krizhevsky et al. (2009) multiclass classification problem. Both MNIST and Fashion-MNIST datasets are grey-scale images of $28 \times 28$ pixels. The CIFAR-10 dataset consists of colored images of $32 \times 32$ pixels. Results are shown in Table 1 [3]. For all three datasets, NOVI outperforms other three methods with significant less training time and iterations. We also perform experiments using three UCI classification datasets and present results in App. D.1.

---

[3]For 3-layer DSVI and SGHMC models, since they have not yet released the corresponding code to reproduce it, we only test the training time and report its iterations according to the original paper.

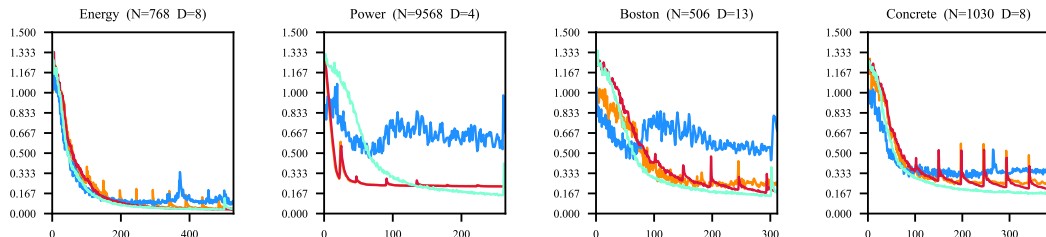

Figure 2: The mean RMSE comparison of NOVI (test: orange, train: red) with Monte Carlo log-likelihood maximization method (test: blue, train: cyan) using 2-layer DGP model on four UCI regression datasets.

### 7.3 COMPUTATIONAL COMPLEXITY

We have compared training efficiency with other three methods on a single GPU [4] using Energy dataset. Results are shown in Table 2. It can be seen that when our model takes less time per iteration than DSVI and SGHMC. Morever, we only need less than one-tenth of the number of iterations to converge compared with the other three methods. Also, as shown in Table 1, for high-dimensional image datasets, NOVI also requires significant less training time and iterations to converge, which shows that the proposed method is scalable to larger datasets. Comparison about numbers of inducing points can be seen in App. D.4.

### 7.4 ABLATION STUDY

To demonstrate the effectiveness of NOVI, we directly maximize log-likelihood with random initialized $\mathbf{U}$ and hyperparameters $\nu$ and compare it with our method using 2-layer DGP model. Results are shown in Figure 2. For all datasets, it can be observed that NOVI yields lower test RMSE and higher train RMSE, hence indicating that our optimization method reduces overfitting. Although the loss fluctuation occurs during the training of our method, it is caused by the unique adversarial training and converges to a stable value after only several hundred iterations. Additional results for ablation study on classification datasets can be seen in App. D.2.

## 8 CONCLUSION

This paper presented a novel NOVI framework to incorporate Stein Discrepency with DGPs that can effectively model a non-Gaussian and hierarchy-related posterior, thus further enhancing the flexibility of DGP models. To achieve this, we generate inducing variables from a neural generator and optimize it jointly with variational parameters through adversarial training. Furthermore, we theoretically demonstrate that the bias raised by our method can be bounded by Fisher divergence, which provides a clear and concise tool to optimize the neural generator. Empirical evaluation shows that NOVI outperforms the state-of-art approximation methods both in regression and classification. The proposed method also requires significant less training time and iterations to converge, which shows that NOVI is more scalable to larger datasets. Due to the nature of adversarial training, NOVI will inevitably encounter fluctuations in loss during training, causing certain difficulties in optimization, but experimental results show that the fluctuations are greatly alleviated at convergence point. Future work includes implementing convolution structure to better extract features from images and utilizing Neural Architecture Search (NAS) method to obtain a more suitable network architecture for practical applications.

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
