# OpenReview forum: "Neural Operator Variational Inference based on Regularized Stein Discrepancy for Deep Gaussian Processes"
_ICLR.cc/2023/Conference — Submitted to ICLR 2023_

### Official Review · Reviewer_GwnC · 2022-10-23

**Confidence:** 4
**Correctness:** 3
**Technical Novelty And Significance:** 1
**Empirical Novelty And Significance:** 2
**Recommendation:** 3

**Clarity, Quality, Novelty And Reproducibility:**

Novelty & Quality:

As stated in the paper summary, my take of this paper is that its main contribution is a re-parameterization of the surrogate posterior q(.) in the formulation of OVI -- this is highlighted in Section 4.1 but this is exactly the same as the re-parameterization trick.

Moreover, such re-parameterization has been indeed incorporated in the original KSD work of (Liu & Wang, 2016) -- see the text after Eq. (4) in https://arxiv.org/pdf/1608.04471.pdf

As such, NOVI is not a new variant of OVI but it is instead an application to DGP with the choice of re-parameterized surrogate put on the distribution of inducing inputs.

I view this as an empirical report advocating for the use of OVI in DGP but even in this regard, this is still lacking in comparison with other related works. The field of DGP contains more than the two baselines the authors compare with.

Experiments:

For example, deep GP learning & inference can also be made via expectation propagation -- https://arxiv.org/abs/1602.04133; the Bayesian treatment of the inducing inputs has also been explored -- see http://web.mit.edu/~jaillet/www/general/neurips2019_main.pdf

For the latter work, isn't the part below Eq. (3) the same as what the authors proposed in 4.1 (which is the only new content added to OVI) & shouldn't it be compared with given that it is more recent and also compared with the two baselines used in this paper on almost the same set of datasets?

There is also another work in the line of Stein Discrepancy: https://arxiv.org/pdf/2009.12141.pdf which is also much more recent.

Note that I wouldn't request comparison with algorithms of other families such as the above if the core of this paper is about a new insight on expanding OVI (even if it is incremental) but in this case, it is a direct application of OVI to DGP and this entire work reads to me as an advocation of using OVI for learning DGP. This is also a fine angle but it has to be compared with other families of algorithms.

Suggestion for Future Improvement:

In fact, given the core components of this paper, I'd suggest the authors to at least revise the stance of this paper towards an empirical comparison of inference algorithms for DGPs and analysis of their pros and cons from both theoretical & experimental perspectives. Then, even if there is no particularly new algorithmic component, the insights from such experiments would still be useful to the community which is meritable. Otherwise, presenting it as an extension of OVI is not convincing.

--

There are also interesting theoretical facts established by this paper but they do not contribute to the algorithm designs. I believe these  are good-to-know facts but do not impact the algorithm design or shed important insights on its numerical behavior.

Clarity:

The paper is clear on the high-level flow but there are a lot of typos, e.g. the 3rd last sentence of the abstract is only half a sentence; "distrbution" -- before Eq. (7); "objictive" -- 2nd sentence after Eq. (12)

There is also particularly problematic typo -- in 4.2 the authors stated that "In Section 3, we have developed OVI" which is not true given the cited reference in Section 3. An appropriate statement would be "we have reviewed OVI".

Fig 1. can be further improved in quality and clarity, e.g. annotations and labels could be added to make it clearer what the notations stand for (even though those are mentioned in the text narrative) -- the caption can also be more informative.

Citations for the mean-field assumption should be mentioned as soon as the assumption was mentioned in the Introduction. Also, in the first paragraph of the Introduction, there is a statement about the constructed probability distribution (in GP) is far from posterior contraction -- please do consider citing references that explore & demonstrate this.

Below Eq. (11): it is not immediately clear why the authors mention the non-integrability of neural network. It seems to be a condition that needs to be enforced by including a regularizer on the space of neural network generators; so please make it clear the connection between this statement & the construction in Eq. (12) -- why such regularizer helps encourage the neural net generator to meet the L2 constraint.

Reproducibility:

The code is released and the presentation of the algorithm is clear so I believe the reproducibility of this work.

**Strength And Weaknesses:**

Strengths:

+ This paper is well-organized and has a good review section on important background of deep GP, Stein Discrepancy and SD-based inference techniques for generic deep Bayesian models such as KSD and OVI

+ There are theoretical results characterizing the behavior of the optimal loss when the Stein Identity is satisfied; and an upper bound on the bias of the DGP predictor using the model learned via OVI.

Weaknesses:

- The contribution stance of this paper is somewhat problematic. It is in fact not clear to me what are the new algorithmic insights that the proposed algorithm (NOVI) provides in addition to what were found in OVI and KSD -- I will elaborate more on this in the Novelty section.

- There is a lack of comparison with other relevant inference work for deep GP.

- Despite the well-organized structure, this paper has quite a no. of typos -- some of which is quite problematic as it changes the semantic of the sentence into a contribution claim of an existing work -- more on this later.


**Summary Of The Paper:**

This paper describes a learning and inference algorithm for deep Gaussian processes based on operator variational inference (OVI).

Particularly, OVI (Ranganath et al., 2016, Hu et al. 2018, Grathwohl et al. 2020) generalizes Kernelized Stein Discrepancy (KSD) to situations where the RKHS space the smooth function (of the Langevin-Stein Operator) is replaced by the L2 space. Key to both OVI and KSD is the computation of the Stein Discrepancy (SD) which is parameterized by both the smooth operator and a surrogate of the true posterior that we wish to sample from.

In KSD, we can solve for SD in closed form which eases the optimization of the surrogate. In OVI, we parameterize the smooth function by a deep neural net and the SD can be estimated via sampling with re-parameterization & solving a minimax optimization task. Both OVI & SGD can be considered generic inference strategy for deep generative learning (including DGP). This paper extends OVI by also parameterizing the posterior with a neural net generator, which appears to be using the re-parameterization trick (Kingma & Welling, 2013).

This is demonstrated on deep Gaussian processes applied to 4 UCI regression datasets + 3 classification benchmark datasets (MNIST, Fashion-MNIST and CIFAR-10) showing marginal improvement over two baselines DSVI (Salimbeni & Deisenroth, 2017) and SGHMC (Havasi et al., 2018).

**Summary Of The Review:**

Currently, my take of this paper is that it is an application of OVI to learning DGP. Despite the claim that it develops NOVI which is an extension of OVI via the neural net generator of the posterior in Section 4.1, this is not new. First, it is on high-level the same as re-parameterization which has been incorporated into KSD (a preliminary version of SD-based VI which OVI was built on). Second, even if we want to discuss it specifically in the context of stochastic inducing input, the same perspective has also been explored in a recent NeurIPS-19 paper -- http://web.mit.edu/~jaillet/www/general/neurips2019_main.pdf.

Given that no new algorithmic insight was introduced, I do not believe the current position of this paper merits an acceptance. But I do suggest another angle for this work (see my specific suggestions above) which is more meritable but that requires more extensive experiments, involving comparison with other families of inference algorithm for DGPs as well as detailed analysis of their modeling pros & cons (demonstrating explicitly deficiencies of existing approaches in the specific context of DGP; and showing how those could be remedied by using OVI).

---

These are my preliminary assessment of this paper. I am of course happy to discuss if the authors disagree.

---

> ### Author Response · Authors · 2022-11-16
> **Response to Reviewer Gwnc**
>
> Thanks for your constructive comments and support.
>
> Q1: The contribution stance of this paper is somewhat problematic. It is in fact not clear to me what are the new algorithmic insights that the proposed algorithm (NOVI) provides in addition to what were found in OVI and KSD:There is a lack of comparison with other relevant inference work for deep GP.
>
> R1: IPVI is a variant of its baseline DSVI while our method NOVI is a generalization of OVI framework for DGP field. We have given a detailed discussion in 'Response to all reviewers' for this issue. Please head there for details.
>
> Q2: Despite the well-organized structure, this paper has quite a no. of typos -- ...
>
> R2: Thanks. We have modified it and re-submitted to “Rebuttal Revision”.
>
>
> Q3: For example, deep GP learning $\&$ inference can also be made via expectation propagation ...the Bayesian treatment of the inducing inputs has also been explored...shouldn't it be compared with given that it is more recent and also compared with the two baselines used in this paper on almost the same set of datasets? There is also another work ...
>
> R3: 'Expectation propagation' and 'IPVI DGP (NeurIPS 2019)' are both based on KL divergence optimization while our method is based on OVI and RSD. It appears that the reparameterization trick of stochastic inducing input of 'IPVI DGP (NeurIPS 2019)' is the same, but essentially we used different approaches. For example, we do not have the concept of 'density ratio, which is derived from the KL regular term of
> DSVI. IPVI runs under the framework of DSVI, but our approach proposes a new DGP inference framework and provides better accuracy and faster convergence. And [pinder2020stein] did not dissuss 'Deep Gaussian processes’ and 'inducing points'. Morever, we have added a section ’Related Work’ and a detailed comparison with IPVI DGP (NeurIPS 2019) in our ’Experiments’. We re-submitted it to “Rebuttal Revision”. Please head there for details.
>
>
> Q4: There are also interesting theoretical facts established by this paper but they do not contribute to the algorithm designs. I believe these are good-to-know facts but do not impact the algorithm design or shed important insights on its numerical behavior.
>
> R4: We have added an explanatory paragraph to Theorem 2 and Theorem 3. Please head there for details.
>
>
> Q5: The paper is clear on the high-level flow but there are a lot of typos...There is also particularly problematic typo...
>
> R5: Thanks. We have modified it and re-submitted to “Rebuttal Revision”
>
> Q6: Fig 1. can be further improved in quality and clarity, e.g. annotations and labels could be added to make it clearer what the notations stand for (even though those are mentioned in the text narrative) -- the caption can also be more informative.
>
> R6: Given that most of the methods have already been explained in the main text, we have removed Figure 1 to save space.
>
>
> Q7: Citations for the mean-field assumption should be mentioned as soon as the assumption was mentioned in the Introduction. Also, ...that explore $\&$ demonstrate this.
>
> R7: Thanks. We have modified it and re-submitted to “Rebuttal Revision”.
>
> Q8: Below Eq. (11): it is not immediately clear why the authors mention the non-integrability of neural network. It seems to be ... why such regularizer helps encourage the neural net generator to meet the L2 constraint.
>
> R8: Stein’s Identity (in Appendix. C) requires the norm of $\phi$ to be bounded outside the support set of $p(x)$, so we add $L_2$ constraint on $\phi$ to meet this requirement. We have modified it and re-submitted to “Rebuttal Revision”.

---

### Official Review · Reviewer_vt6c · 2022-10-24

**Confidence:** 4
**Correctness:** 3
**Technical Novelty And Significance:** 2
**Empirical Novelty And Significance:** 2
**Recommendation:** 3

**Clarity, Quality, Novelty And Reproducibility:**


# Clarity

The paper is well-written, covering previous contributions and the derivations needed to understand the proposed method.

Please, check for typos. Some examples are:
  * "framwork" (first contribution bullet point, page 2)
  * "objictive" (three lines below Eq. 12)
  * "gemoetry" (second line of Section 4.2)


# Quality

The paper is of good quality, although some details should be polished to make justice to the amount of work behind this contribution. More specifically, providing better plots and graphical models would help, as well as taking care of the mishaps of the tabulated data in the supplementary material.

# Novelty

The ingredients for the method already exist before the proposal, so the contribution is mainly centered on a particular combination of these well-known concepts in a new manner that results beneficial.


# Reproducibility

Code is already publicly available and the text is clear. Without attempting to reproduce it myself, I deem the contribution reproducible.


**Strength And Weaknesses:**

# Strengths:

* The method is an interesting addition to the literature in Deep Gaussian Processes, a promising line of research that has layed out very interesting results for the past few years.

* The approach is fairly easy to understand. Even though there are important details that must be covered in detail, for the most part it is pretty straight-forward.

* The authors provide extensive experiments, accounting also for the scalability of the method and its applicability in different problems and datasets. This strenghtens the submission, as well as the convergence guarantees and the extra information provided in the supplementary material.

* The paper is mostly well written and easily understandable, covering the most important points on previous works and providing a good description of the method's development.


# Weaknesses:

* My main concern here is the novelty of the method, which I deem to be a bit lacking. Overall, I see the contribution as interesting to the community, and also potentially very useful. However, as far as I understand it, this method results from the combination of previously well-known previous contributions in a very direct manner. I may be mistaken, and in that case I am entirely open to change my opinion on this matter, but as it is I think the contribution may lack sufficient novelty.

* I also suggest the authors to emphasize in a stronger note the differences between this method and previously existing contributions. This may help with the previous point as well.

* The UCI experiments could be much more robust if other metrics were reported asides from the RMSE. Common choices are the negative log-likelihood or the continously ranked probability score, as a proper scoring rule. I suggest the authors to include some other metrics that help better assess the quality of the predictive distribution, since RMSE does not necessarily provide that information.

* I would appreciate a more extensive experiment with bigger datasets than Energy (e.g. Taxi, Airlines or others similar to these ones). Also, it would help if the authors shed some light on the expected performance of the method in cases with high dimensionality.

* (_minor_) The presentation of the tabulated results in the supplementary material should be revised.

* (_minor_) The "Related Work" section seems limited, since there are several other approximate inference methods strongly related to this one. As an example, the NOVI setup proposed in section 4.2  strongly ressembles the construct made in articles like [5,6] that deal with implicit distributions for approximate inference. Moreover, formulation in the equation of section 4.1 makes me wish there was a discussion between the relation between these methods and function-space inference with implicit stochastic processes, since the two approaches actually have a lot in common. As suggestions, I see strong relationship with works such as [1,2,3,4].

* (_minor_) Figure 1 should be re-thought, its quality leaves some improvement to be desired to help understanding the content of the graphical models.


## Other comments:

* The language employed in section 4.2 makes it seem sometimes that the authors contribute here to the development of previous techniques, e.g. OVI (Ranganath et al. 2016). I suggest the authors to be clarify the language here to state what contributions are genuinely theirs.

* Please, clarify further the statement before Eq. 14 regarding the fact that the two distributions must be "equivalent"



## References:


[1] Ma, C., Li, Y., and Hernández-Lobato, J. M. (2019). “Variational implicit processes”.
In: International Conference on Machine Learning, pp. 4222–4233.

[2] Sun, S., Zhang, G., Shi, J., and Grosse, R. (2019). “Functional variational Bayesian neural networks”. In: International Conference on Learning Representations.

[3] Rodrı́guez Santana, S., Zaldivar, B., & Hernandez-Lobato, D. (2022). Function-space Inference with Sparse Implicit Processes. In International Conference on Machine Learning (pp. 18723-18740). PMLR.

[4] Ma, C., & Hernández-Lobato, J. M. (2021). Functional variational inference based on stochastic process generators. Advances in Neural Information Processing Systems, 34, 21795-21807.

[5] Mescheder, Lars, Sebastian Nowozin, and Andreas Geiger. "Adversarial variational bayes: Unifying variational autoencoders and generative adversarial networks." International Conference on Machine Learning. PMLR, 2017.

[6] Santana, S. R., & Hernández-Lobato, D. (2022). Adversarial α-divergence minimization for Bayesian approximate inference. Neurocomputing, 471, 260-274.


**Summary Of The Paper:**

In this paper, the authors propose a novel framework to conduct approximate variational inference in deep Gaussian processes. It is based on minimizing the regularized Stein discrepancy between the approximate distribution and the true posterior through the newly proposed Neural Operator Variational Inference algorithm for DGPs, which iteratively updates a generator and the associated discriminator to estimate more accurately the target discrepancy. This leads to a better final approximation than the ones obtained through the regular mean-field Gaussian assumption usually employed in DGPs. Authors support this claimg by extensive regression experiments, in which they show also that the method presents important properties such as robustness to overfitting.

**Summary Of The Review:**


The contribution seems correct and interesting to the research community both due to its implications in DGPs and VI. However, it may lack sufficient novelty to merit a higher score. The authors should make an effort to highlight in a stronger fashion how does this contribution differenciate from previous works. Some extensions to the current experimental setup could help the case here as well (it is fine already, but some simple additions would make it even stronger).

---

> ### Author Response · Authors · 2022-11-16
> **Response to Reviewer vt6c**
>
> Thanks for your constructive comments and support.
>
> Q1: My main concern here is the novelty of the method, which I deem to be a bit lacking. Overall, ....may lack sufficient novelty.I also suggest the authors to emphasize in a stronger note the differences between this method and previously existing contributions. This may help with the previous point as well.
>
> R1: IPVI is a variant of its baseline DSVI while our method NOVI is a generalization of OVI framework for DGP field. We have given a detailed discussion in 'Response to all reviewers' for this issue. Please head there for details.
>
> Q2: The UCI experiments could be much more robust if other metrics were reported asides from the RMSE. Common choices ... RMSE does not necessarily provide that information.
>
> R2: The essence is much the same. RMSE seeks the predicted sampling average, while NLL seeks the actual mean and noise variance, but the forward process also requires sampling. We will add more metrics in the next version.
>
> Q3: I would appreciate a more extensive experiment with bigger datasets than Energy (e.g. Taxi, Airlines or others similar to these ones). Also, it would help if the authors shed some light on the expected performance of the method in cases with high dimensionality.
>
> R3: As we mentioned in the supplementary material, the generator was constructed using two fully connected layers (which also performed well even in image datasets). Direct training and sampling schedule offers a scalable approach for DGPs to extend to bigger datasets. For bigger datasets (with more than 100,000 or a million data points), it can be trained by subsampling into minibatch, and the size of the batch is generally limited by computer storage. We have also conducted some classification experiments on high-dimensional image datasets such as MNIST and CIFAR-10. We will add more high-dimensional regression datasets (e.g., Airlines) in the next version.
>
> Q4: (minor) The presentation of the tabulated results in the supplementary material should be revised.
>
> R4: Thanks. We have modified it and re-submitted to ``Rebuttal Revision''.
>
> Q5: (minor) The "Related Work" section seems limited, since there are several other approximate inference methods strongly related to this one. As an example, ...
>
> R5: We have grouped the six articles you mentioned into one category in 'Related Work', i.e., implicit distributions variational inference (IPVI). Such methods are essentially different from ours, although the posterior is generated in the same way. The main issue of IPVI is density ratio estimation, while we have no such concept. IPVI in general is based on the original VI framework, while we are based on Operator VI.
>
> Q6: (minor) Figure 1 should be re-thought, its quality leaves some improvement to be desired to help understanding the content of the graphical models.
>
> R6: We have removed it since the context in the main text has provided enough details to elaborate.
>
> Q7: The language employed in section 4.2 makes it seem sometimes that the authors contribute here to the development of previous techniques, e.g. OVI (Ranganath et al. 2016). I suggest the authors to be clarify the language here to state what contributions are genuinely theirs.
>
> R7: Sorry for the typo. An appropriate statement should be "we have reviewed OVI". Our contributions have been represented in 'Introduction' and relation between our method and OVI has been discussed in 'Related Work'.
>
> Q8: Please, clarify further the statement before Eq. 14 regarding the fact that the two distributions must be "equivalent"
>
> R8: ``Equivalent'' means two distributions are almost equal everywhere with respective to probability measure.

---

### Official Review · Reviewer_C7ZF · 2022-10-25

**Confidence:** 5
**Correctness:** 3
**Technical Novelty And Significance:** 2
**Empirical Novelty And Significance:** 2
**Recommendation:** 3

**Clarity, Quality, Novelty And Reproducibility:**


Clarity: The paper is well-written and relatively easy to follow.
Quality & originality: The contribution of this paper is relatively incremental
Reproducibility: The code is provided with the paper, I believe the reproducibility of this work.

**Strength And Weaknesses:**

In fact, this paper demonstrates a high resemblance to Implicit Posterior Variational Inference for Deep Gaussian Processes [1]. The difference is that this work decides to minimize the Stein discrepancy while the work of [1] minimizes KL divergence. The design of the generator and discriminator, the iterative optimization procedure, and even the design to represent the inducing variables.

These two methods are not even compared in this paper, given all these resemblances. From my point of view, this comparison is compulsory. I am happy to raise my score if the authors can provide convincing reasons.

In the context of the NOVI algorithm, this paper is still far from satisfactory. The convergence guarantee is provided in Theorem 2 and Theorem 3 and yet no analysis or illustrations to prove this theorem. How would this method perform given a known true posterior distribution? This kind of synthetic experiment will make the claims more convincing than RMSE or classification accuracy.


[1] https://arxiv.org/abs/1910.11998

**Summary Of The Paper:**

This paper proposes a new method for inference of the intractable posterior for deep Gaussian processes (DGPs). The idea is to minimize the Stein discrepancy between the approximated posterior distribution which is demonstrated by a generator and the true posterior distribution. The algorithm utilizes a discriminator to estimate the discrepancy between two distributions. It is claimed that a better approximation to the true posterior is recovered. To support the claim, extensive regression and classification experiments are conducted.

**Summary Of The Review:**

Given that the novelty and contribution of this paper are relatively incremental, I am inclined to reject this paper. Moreover, I would like to suggest the authors polish this paper in the direction of comparison of different inference methods for DGPs, the pros, and cons for each.

---

> ### Author Response · Authors · 2022-11-16
> **Response to Reviewer C7ZF**
>
> Thanks for your constructive comments and support.
>
> Q1: In fact, this paper demonstrates a high resemblance to Implicit Posterior Variational Inference for Deep Gaussian Processes [1]. The difference is ....
> ... are not even compared in this paper, ....
>
> R1: IPVI is a variant of its baseline DSVI while our method NOVI is a generalization of OVI framework for DGP field. We have given a detailed discussion in 'Response to all reviewers' for this issue. Please head there for details.
>
> Q2: In the context of the NOVI algorithm, this paper is still far from satisfactory. The convergence guarantee is provided in Theorem 2 and Theorem 3 and yet no analysis or illustrations to prove this theorem. How would this method perform given a known true posterior distribution?
>
> R2: We have proved Theorem 2 and Theorem 3 in Supplementary materials (in Appendix.C). Please head there for details. Additionally, We have made an explanatory paragraph to Theorem 2 and Theorem 3. The experiments we performed for regression and classification were derived from baseline DSVI DGP and other DGP methods. We will add some experiments to test the expressiveness of our method on some known true posterior distributions in the next version.

---

### Official Review · Reviewer_wts5 · 2022-10-26

**Confidence:** 3
**Correctness:** 2
**Technical Novelty And Significance:** 3
**Empirical Novelty And Significance:** 3
**Recommendation:** 5

**Clarity, Quality, Novelty And Reproducibility:**

The paper presents an interesting, novel approach.  The writing and exposition could certainly be made clearer.  Theorem 1 is presented a theorem when in reality it is more like a proposed scheme for approximating gradients.

**Strength And Weaknesses:**

Strengths:
* The approach is interesting, and the ability to learn complex posteriors in the DGP framework is promising.

Weaknesses:
* (minor) there are a number of typos throughout the manuscript. This did not interfere with my understanding, but the paper could use some thorough copy eding.   For example there are numerous citation typos (particularly missing spaces before citations in the text).  Also, e.g., "In particular, according to Bayesian formula" --> "In particular, according to Bayes' Rule";  "a quick introduction to these concepts that forms the foundation of our method" --> "a quick introduction to these concepts that form the foundation of our method"; and so on.
* The theorems assume an infinitely expressive neural network architecture, but the role of network architecture on the empirical results is relatively unexplored (e.g., only number of layers, but not number of hidden units, convolutional architectures, etc...)
* In table 1 in the appendix, why does the runtime increase only slightly with the number of inducing points?  If the runtime increases so slightly and accuracy increases substantially, then why not use substantially more inducing points.  E.g., spending an additional 0.011s (presumably per iteration?) on Concrete to go from 50 to 400 inducing points, but going from an RMSE of 0.28 to 0.19 seems like a very good tradeoff.  Why not go on to more inducing points?  When does accuracy plateau or when does the runtime become infeasible?
* In equation (16), a non-random quantity on the left hand side is claimed to be equal to a random quantity on the right hand side.  This equality does not even hold in expectation (i.e., the gradient will be biased) as can be seen by applying Jensen's inequality to Equation (6) in the appendix.  And the authors are certainly aware that this is not a strict equality given the $\approx$ in Equation (6) in the supplement.  As such, I think that calling this a "theorem" with a strict equality is somewhat disingenous.

**Summary Of The Paper:**

This paper proposes a new method for approximating the posterior of deep gaussian processes, to allow for non-Gaussian posteriors.  The main idea is to train a generator network that approximates the posterior, while training a discriminator that learns a regularized version of the Stein discrepancy.  They also optimize hyperparameters and inducing point locations via SGD.  The authors also prove some theorems in the (unrealistic, but still interesting) regime of having infinitely expressive neural networks than can be perfectly optimized.

**Summary Of The Review:**

The paper presents an interesting approach, but the key assumptions of its theorems are not met in practice (infinitely expressive neural networks, perfect optimization).  The impact of neural network architecture in particular, is not sufficiently explored.  I also found some parts of the paper to be presented in a way to appear rigorous, but are actually sort of pseudo-rigorous, at the expense of both clarity and accuracy (e.g., Theorem 1; repeated claims that neural networks with a finite number of hidden units are expressive enough to model any distribution).

---

> ### Author Response · Authors · 2022-11-16
> **Response to Reviewer  wts5**
>
> Thanks for your constructive comments and support.
>
> Q1:(minor) there are a number of typos throughout the manuscript. ... and so on.
>
> R1: Thanks. We have modified it and re-submitted to ``Rebuttal Revision''.
>
> Q2: The theorems assume an infinitely expressive neural network architecture, but ...
>
> R2: We use two fully connected layers to construct the generator and present the architecture in Figure 3 of the Supplementary materials. We delete the sentence of this assumption since it does not affect the validity of the theorem. By Cauchy-Schwarz inequality, our training schedule is essentially equivalent to minimizing Fisher divergence if the condition 'training the generator with the optimal discriminator' holds. Moreover, since the proposed method is more robust, NOVI relies less on the flexibility of the neural network and reaches convergence faster than previous methods (detailed in Official Comment: Response to all reviewers)
>
> Q3: In table 1 in the appendix, why does the runtime increase only slightly with the number of inducing points? ...infeasible?
>
> R3: For low-dimensional datasets, our method provides an effective way to model the posterior distribution and generate it using neural network. Keeping the number of inducing points the same is actually for a fair comparison with other methods. We will add more inducing points per layer to discuss its potential expressiveness in the next version.
>
> Q4: In equation (16), a non-random quantity on the left hand side is claimed to be equal to a random quantity on the right hand side. ... is somewhat disingenous.
>
> R4: Although not strictly equal, the equation holds in the expectation sense, i.e., by the Law of Large Number, a sufficiently large sample size S allows the right side of the equation to converge to the left side. The language in Theorem is ' The score function can be evaluated by Monte Carlo
> sampling', which does not mean strictly equal. For the sake of rigorousness, we have changed the 'equal sign' to an 'approximate equal sign'.

---

> > ### Comment · Reviewer_wts5 · 2022-12-09
> > **Thank you for the response**
> >
> > Thank you very much for the response.  Having read the other reviews, I stand by my initial score, but thank you very much for responding and updating the paper.

---

### Author Response · Authors · 2022-11-16
**Response to all reviewers**

 We would like to highlight here the main contributions  and the main differences between our approach and recent advanced [mescheder2017adversarial] and [yu2019implicit], as well as our advantages.

The major differences are shown as follows:

Motivation: While IPVI aims to find the exact density ratio in the framework of DSVI, our method aim to  propose a new DGP inference framework.

Method: a): In the case of neural networks with limited flexibility, NOVI is designed with a better discriminator than IPVI. As a variant of vanilla VI, IPVI purely replaces the regular term after the expectation likelihood term in the ELBO with a neural network discriminator $T(x,z)$, see Equation (1) below, essentially performing density ratio estimation.
$$
    L_{IPVI}= E_{z \sim q_{\psi}}[\log p_{\theta}(x \mid z)-T(x, z)] (1)
$$
where
$$
    T=argmax E_{q_{\psi}(z \mid x)} \log \sigma(T(x, z)) +E_{p_\theta(z)} \log (1-\sigma(T(x, z)))(2)
$$
The problem lies in the fact that this  discriminator is solved by constructing a  functional extremum problem. In each iter, the optimization of the discriminator needs to be globally optimal (hard constraint).If the discriminator is not flexible enough, which results Equation (2) being suboptimal, then (1) cannot be called a 'reasonable' ELBO and it will occur the same overfitting phenomenon, as we have presented in Section 7.3. In contrast, NOVI skips the strongly constrained density ratio estimation and the intractable posterior log pdf, updates the parameters in the gradient domain (i.e., uses the score function), which, by imposing soft constraints, not only breaks the DSVI mean field assumptions but also avoids possible problems caused by the suboptimal discriminator like IPVI. Specially, NOVI derives its discriminator from  Stein discrepancy (3), where the 'sup' is defined by the desire for this distance to be 'strongest'. Thus, even if the discriminant is suboptimal, it can be optimized quite precisely, since Stein's identity (4) for any mild function $\phi$ (non-infinite on support sets of p) holds, i.e. the function domain F can be defined as any set of mild smooth functions. At this point,  F is redefined in the suboptimal set of the discriminator.
$$S(q, p) = sup_{\phi  \in F}E_{x\sim q}[A_p {\phi}(x)] (3)$$where$$A_p \phi(x) = \nabla\log p(x)^{T} \phi(x)+Tr(\nabla {\phi}(x))$$
$$ E_{x\sim p}[A_p {\phi}(x)]=0(4)$$
b): Since the discriminator does not need to be globally optimal, the number of iteration steps is saved, thus greatly speeding up the convergence speed and training time.

c): Step back and assume that the discriminator can reach the global optimum, i.e., IPVI finds the exact density ratio estimation, which is only equivalent to optimizing the ELBO induced by KL divergence (It is known that KL can be small even when the approximate point estimations and uncertainties are arbitrarily far from the exact values [huggins 2018 practical]).
If NOVI's discriminator achieves global optimality, by contrast, it is equivalent to optimizing Fisher divergence (Theorem 2), which is stronger than most of the well-known divergence (e.g., KL, Wasserstein distance) and provides better moment estimations than KL, reducing the burden on the generator at another hand (i.e., fewer iterative steps), thus accelerating convergenc

d):Compared with IPVI, there is no need to sample from the prior and extra noise is avoided.

Performance: Compared with IPVI, NOVI achieves significant improvement in performance (robustness, accuracy, number of iteration steps and convergence rat).Our additional experiments are shown in the tables below and the results confirm our idea.

UCI datasets  RMSE for DGP2:

 Dataset   NOVI |IPVI

boston 0.20(0.01) | 0.35(0.06)

energy 0.04(0.00) |0.13(0.01)

concrete 0.24(0.00) |0.32(0.02)

kin8nm 0.24(0.00) |0.25(0.01)

protein 0.67(0.00) |0.68(0.01)

Image classification accuracy for DGP4:

MINIST 98.01|  97.80

Fashion-MN 89.15 | 88.90

Cifar-10 53.42| 53.27

Average number of iteration steps:

DatasetNOVI| IPVI

UCI regression 500  | 20K

Image classification 10K | 20K

The main contribution of this paper is the development and advocacy of a scalable NOVI-based DGP inference framework. Despite the benefits of OVI over VI and its variants in terms of faster convergence and greater accuracy mentioned above, it is surprising that in the literature such a  framework has not been established and applied in specific scenarios for DGP, due to sampling and calculation difficulties and many algorithmic details need to be designed and deliberated carefully. We design a scalable algorithm for DGP that includes the generation of the inducing point approximation posterior, MC evaluation of the score function, the Hutchinson estimation of the gradient term, and joint optimization of inducing points and hyperparameter point estimation and illustrate its advantages and limitations.
Moreover, a theoretical and experimental demonstration for convergence is provided.

---

### Decision · Program_Chairs · 2023-01-20

**Decision:**

Reject

**Justification For Why Not Higher Score:**

* Lack of novelty
* Lack of experimental comparison to competing approaches

**Justification For Why Not Lower Score:**

N/A

**Metareview: Summary, Strengths And Weaknesses:**

This work proposes an approximate posterior inference algorithm for deep Gaussian processes. A sampleable approximate posterior parameterized by a neural network minimizes a regularized Stein discrepancy objective between itself and the true posterior distribution. Experiments are conducted on regression and classification problems.

The main concern, as shared by Reviewer vt6c and GwnC, is its lack of novelty. The main contribution over, for example, operator variational inference and kernel stein discrepancy, is its use of a neural parameterization and toward the specific model of deep Gaussian processes. This feels too incremental (and in fact OVI already included experiments with a neural generator). A thorough experimental evaluation could also be one of the paper's main contributions, but unfortunately there is also quite a lack of experimental comparison to other DGP inference methods.

All reviewers leaned toward reject, and I agree with their opinion.